# Visceral Leishmaniasis in Immunocompetent Hosts in Brescia: A Case Series and Analysis of Cytokine Cascade

**DOI:** 10.3390/microorganisms12020394

**Published:** 2024-02-16

**Authors:** Alice Mulè, Verena Crosato, Douglas Byron Kuhns, Luisa Lorenzi, Claudia Chirico, Giovanni Maifredi, Luigi D. Notarangelo, Francesco Castelli, Lina R. Tomasoni

**Affiliations:** 1Unit of Infectious and Tropical Diseases, Department of Clinical and Experimental Sciences, University of Brescia and ASST Spedali Civili di Brescia, 25123 Brescia, Italy; v.crosato001@unibs.it (V.C.); francesco.castelli@unibs.it (F.C.); 2Neutrophil Monitoring Laboratory, Applied/Developmental Research Directorate, Frederick National Laboratory for Cancer Research, Frederick, MD 20701, USA; dkuhns@mail.nih.gov; 3Pathology Unit, Department of Molecular and Translational Medicine, University of Brescia, 25123 Brescia, Italy; luisa.lorenzi@unibs.it; 4Department of Hygiene and Health Prevention, Health Protection Agency of Brescia (ATS Brescia), 25124 Brescia, Italy; 5Epidemiology Unit, Health Protection Agency of Brescia (ATS Brescia), 25124 Brescia, Italy; giovanni.maifredi@ats-brescia.it; 6Laboratory of Clinical Immunology and Microbiology, Division of Intramural Research, NIAID, NIH, Bethesda, MD 20852, USA; luigi.notarangelo2@nih.gov; 7Unit of Infectious and Tropical Diseases, ASST Spedali Civili di Brescia, 25123 Brescia, Italy; linatomasoni@yahoo.it

**Keywords:** *Leishmania*, immunocompetent host, visceral leishmaniasis, cytokines, Th1, Th2, IFN-γ, Brescia, parasitic zoonosis, neglected disease

## Abstract

Visceral leishmaniasis (VL) is a parasitic zoonosis caused by *Leishmania* spp. that usually manifests itself in immunocompromised subjects. It is a rare and neglected disease, and it is not endemic in the province of Brescia (Italy). Three cases of human VL occurred in Brescia from October to December 2021 in immunocompetent patients. We evaluated the patients looking for signs of underlying immunodeficiencies and conducted further epidemiological evaluations in the province of Brescia without success. An analysis of the sera levels of the main cytokines involved in the immune response to VL was performed. All patients presented a significant augmentation of CXCL-10, CCL-4, and IL-6. The patients tested during the acute phase showed an elevation of IL-1α, IL-5, IL-10, and IL-12, while in the recovery phase, higher levels of TNF-α and IL-7 were detected. Altogether, a predominant activation of the T-helper-2 pathway emerged during the acute phase of the parasite infection, while the cytokines associated with the T-helper-1 pathway were less represented. This imbalanced immune response to the parasite infection might play a crucial role in the development of VL in immunocompetent patients.

## 1. Introduction

Leishmaniases are a group of vector-borne parasitic zoonoses caused by protozoa belonging to the genus *Leishmania*. The parasite is mostly transmitted to humans by an infected sandfly carrier belonging to the genus *Phlebotomus* or *Lutzomya*. The vectors become infected by biting an animal reservoir (such as dogs or other mammals). In very rare occasions, the parasite may be transmitted by blood transfusions, needle sharing, or during pregnancy [1,2,3,4]. A possible anthroponotic transmission is described for *L. donovani*: when a host presents abundant parasitemia or a post-kala-azar dermal lesion, the infection may be transmitted from human to human through sandflies’ bite (see Figure 1) [2].

The genus *Leishmania* comprises four subgenera: *Leishmania*, *Sauroleishmania*, *Mundinia*, and *Viannia*. They include various species [5]. Molecular techniques based on DNA amplification can be used on different samples to identify the species [6].

Visceral leishmaniasis (VL) of the Old World is mainly caused by *L. donovani* and *L. infantum*, which are obligate intracellular parasites. VL can present as an asymptomatic, mild, severe, and even fatal infection. The heterogeneity of clinical manifestations reflects the complex relationship between the parasite and the host [7]. The main symptoms are fever, fatigue, pancytopenia, and hepatosplenomegaly [8,9,10]. Congenital or acquired immunodeficiencies, such as HIV infection, are related to severe forms of VL [2,11].

Also, the infecting strain of *Leishmania* determines the clinical manifestations of the disease. For example, *L. infantum* and *L. donovani* are more likely to cause VL.

### 1.1. Epidemiology

VL is endemic in 79 countries, where over 1 billion people are at risk of infection. The reported worldwide incidence is 12–13,000 cases/year, but the estimated incidence is about 30–60,000 cases/year due to gaps in surveillance systems and unknown underreporting [12,13]. Nowadays, VL still causes about 8000 years lived with disability annually, and a 9-time higher burden of premature deaths [8,9].

Only 2% of all WHO cases are reported in Europe, where *Leishmania* is endemic in 27 countries, including Italy [12]. The European *LeishMan* network reported 241 VL cases from 2014 to 2019 [14]. *L. infantum* is reported in European-acquired cases [15].

Italy, where VL notification is mandatory, reports less than 100 cases/year [12]. The most afflicted areas in Italy are Sicily, Sardinia, the southern coastal areas, and the northern Tyrrhenian shores [16]. However, since the 90s, the diseases have shown a south-to-north spread, maybe related to the higher HIV infection prevalence in these areas [17]. In 2012–2013, an outbreak of 14 cases of VL was reported in Bologna in Italy, where that many cases had never been presented in the previous 40 years, and only 1 infection occurred in a person living with HIV (PLWH) [18]. In Bologna, the prevalence of *Leishmania* antibodies was demonstrated in 12.5% of blood donors in 2014-5 [19]. Since then, an upsurge of human leishmaniasis has been reported in the Emilia-Romagna region, raising doubts about the role of domestic versus wild animals as reservoirs [11,20,21,22].

Autochthonous canine leishmaniasis (aCL) has been reported for some years in all of northern continental Italy. In Lombardy, a recent survey identified aCL in four provinces, but not in Brescia, where aCL was reported more than ten years ago [23]. In the population of the Health Protection Agency (HPA) of Brescia, only 7 cases of human VL were notified from 2005 to 2021, with an incidence rate of 0.37 cases per million people per year.

### 1.2. Physiopathology

The first target of the promastigotes are neutrophils, the earliest cells recruited to the bite site, resulting in neutrophil apoptosis. Both free parasites and infected apoptotic neutrophils are then taken up by dendritic cells and macrophages, which then spread to visceral organs carrying the amastigotes. As neutrophils are non-suitable host cells for intracellular parasites due to their short lifespan, macrophages and dendritic cells act as appropriate hosts for *Leishmania* [24,25].

The innate immune system and the cytokine cascade have a pivotal role in early response against VL. Interleukin (IL)-12 produced by dendritic cells triggers natural killer (NK) cell activation. NK cells secrete interferon gamma (IFN-γ), inducing a leishmanicidal mechanism that inhibits the parasite spread until a T-cell response is fully activated.

The resolution of infection is associated with the expansion of CD4+T-helper-1 (Th1) cells that secrete IFN-γ and IL-12. Also, CD8+ T-cells have a cytotoxic activity against infected macrophages [26,27]. On the other hand, T-helper-2 (Th2) cells, through secretion of IL-4 and IL-10, inhibit macrophage activation and enhance the susceptibility of the host to the parasite and the development of VL. Severe VL in humans is associated with the production of transforming growth factor (TGF)-β and IL-10, which suppress the Th1 response, and the activation of macrophages by IFN-γ (see Figure 1) [2,28,29].

Anti-*Leishmania* antibodies are produced during VL, but they are not protective.

To date, we do not fully understand why the Th1 response arises in some hosts and not in others [30]. VL is considered an immunodeficiency-related disease, often as an opportunistic infection among PLWH [31]. Nevertheless, in rare cases, even immunocompetent hosts may develop VL [2].

## 2. Case Series

Hereby, we present three VL cases, diagnosed and treated in Spedali Civili of Brescia. Their risk factors for *Leishmania* exposure and VL progression are listed in Table 1.

Case 1: The first patient was an 80-year-old Italian male. He lived in the province of Brescia and reportedly traveled to Greece in 2019. He suffered from nephrolithiasis and arterial hypertension but had no known immunodeficiencies, nor a history of previous *Leishmania* infection or infective diathesis. He owned a dog, who appeared to be healthy and tested negative for *Leishmania*. The patient never acquired SARS-CoV-2 infection and received two shots of the Comirnaty vaccine in March 2021. Since then, he experienced weight loss, fatigue, and night sweats. In June 2021, he underwent a bone marrow biopsy on the suspicion of a primary hemopathy, which produced negative results. Due to the persistence of the same symptoms, he repeated the procedure in October 2021, which demonstrated the presence of *Leishmania* amastigotes in the patient’s bone marrow (see Figure 2).

He was admitted to our Infectious Diseases ward in Brescia on 22 October 2021. Blood tests showed pancytopenia and hypergammaglobulinemia and his physical examination highlighted splenomegaly. During his hospital stay, his first bone marrow biopsy was reevaluated, and amastigotes were identified on that sample as well. Serology tested positive for *Leishmania* spp. antibodies. Real-Time Polymerase Chain Reaction (qPCR) on peripheral blood identified *L. donovani*.

He underwent targeted treatment with liposomal amphotericin B (3 mg/kg IV daily on days 1–5, 14, and 21) and fully recovered. A follow-up visit was performed 3, 6, and 12 months later and no relapse or sequelae were identified.

Case 2: The second patient was a 59-year-old female. She was born in Albania but living in Italy since 2008. She suffered from previous HBV infection, chronic gastritis, arterial hypertension, and hypothyroidism. She had no known immunodeficiency and did not own pets. She had no recollection of being affected by COVID-19, but her serology tested positive for anti-nucleocapsid antibodies. She underwent one shot of the SARS-CoV-2 Janssen vaccine in June 2021. In early August 2021, she began experiencing fever with chills, fatigue, night sweats, and anorexia. Due to the persistence of such symptoms, she performed an abdominal ultrasound, which revealed an isoechogenic lesion of the liver.

Because of these findings, in late December 2021, she was admitted to the Surgical ward on suspicion of a neoplastic disease. During her stay, she presented with fever and fatigue and was also diagnosed with pancytopenia, hypergammaglobulinemia (complicated by two monoclonal gammopathies), and splenomegaly. After an infectious disease consultant evaluation, she was tested for *Leishmania* serology, which returned positive results.

She was transferred to the infectious diseases ward and a bone marrow biopsy revealed amastigotes (see Figure 2). qPCR on peripheral blood detected *L. infantum*.

She was treated with liposomal amphotericin B (3 mg/kg IV daily on days 1–5, 14, and 21). During her hospital stay, she needed blood transfusions. Later, her general condition improved, and she was discharged in January 2022 in good health.

At the 3-month follow-up visit, she presented with mild anemia and fatigue and, therefore, started supplementation therapy with iron and B9 vitamin. Three months later, her blood count was normal, and no more follow-up visits were performed.

Case 3: The last patient was a 24-year-old Italian male. He had no history of chronic diseases. He owned a dog (negative for *Leishmania* spp.) and reported spending his 2021 holidays in Corfu. He received two doses of the SARS-CoV-2 vaccine (Vaxzevria and Comirnaty) in the Spring of 2021. In the same year, he acquired mild SARS-CoV-2 infection.

Since August 2021, he suffered from night sweats, fatigue, and intermittent fever. He presented with pancytopenia, hypergammaglobulinemia (two monoclonal gammopathies), and moderate splenomegaly. Upon the suspicion of a primary hemopathy, he was subjected to a bone marrow biopsy with the finding of numerous amastigotes (see Figure 2).

He was therefore admitted to our infectious diseases ward in December 2021. qPCR on his blood samples tested positive for *L. infantum*. Hematological disorders were ruled out, as were HIV infection or other immunological deficits.

He was treated with liposomal amphotericin B (3 mg/kg IV daily on days 1–5, 14, and 21) and fully recovered. At the 3-month follow-up visit, he presented with a mild low platelet count, which was resolved during the following month without any pharmacological intervention.

## 3. Aim of the Study

Such an increased incidence in a short lapse of time of VL in otherwise healthy people in Brescia is an outlier. Therefore, we chose to conduct further laboratory investigations.

Our main hypotheses were that either the COVID-19 pandemic (or its vaccinations) had an influence on the development of VL, or an unknown epidemic cluster of leishmaniasis was spreading in Brescia. Lastly, another explanation might be that the patients were affected by any kind of previously undiagnosed immunodeficiency.

We collected the patients’ sera to study the cytokine cascade involved in the progression of VL, looking for any alteration that might cause the development of the disease.

## 4. Materials and Methods

All the diagnoses of VL were confirmed by qPCR. The molecular identification was performed by the Laboratory of Microbiology of San Raffaele Hospital in Milan. Starting from peripheral blood, the DNA is extracted and purified through ELITe InGenius (by ELITechGroup). Subsequently, the rotor gene was operated in the qPCR determination, using a homemade kit (primer and probe by Eurofins).

Sera from all three patients were collected and sent to the Laboratory of Clinical Immunology and Microbiology, National Institute of Allergy and Infectious Diseases, National Institutes of Health, Bethesda, MD, USA, where the main cytokines and interleukins responsible for the immune response against *Leishmania* spp. were analyzed.

While sera from Patients 2 and 3 were collected during their hospital stay, serum from Patient 1 was collected after he was discharged. This different timing of sampling probably affected the analysis as serum from Patient 1 was obtained during his recovery time, while the sera from Patients 2 and 3 were collected while the disease was still active.

At first, the cytokine levels of all patients were compared with a pool of 72 healthy volunteers living in a non-endemic area for VL (USA). Subsequently, the cytokines’ patterns of Patient 1 and Patients 2–3 were compared with the healthy controls separately to study the relation between the cytokines’ alterations and the evolution of the disease.

Cytokines considered for the statistical analysis were CCL-2, CCL-3, CCL-4, CCL-11, CCL-13, CCL-17, CCL-22, CCL-26, CXCL-10, GM-CSF, VEGF, LTα, IFNγ, TNF-α, IL-1α, IL-1β, IL-2, IL-4, IL-5, IL-6, IL-7, IL-8, IL-10, IL-12, IL-12p70, IL-13, IL-15, IL-16, and IL-17.

The software used for the statistical analysis were Epi-info (V.7.1) and SPSS (V.28.0.1.1). The analysis was conducted with the ANOVA (*T*-test), Bartlett’s test for inequality of population variances, the Mann–Whitney/Wilcoxon two-sample test (Kruskal–Wallis’s test for two groups), and Student’s *T*-test.

The statistical significance cut-off was considered as a *p*-value inferior to 0.05.

All subjects gave their informed consent for inclusion before they participated in the study. The study was conducted in accordance with the Declaration of Helsinki.

## 5. Results

Both the analysis of all patients’ sera versus healthy controls and the sub-analysis stratified by disease stages reported no significant differences in the patterns of CCL-2, CCL-3, CCL-11, CCL-13, CCL-17, CCL-22, CCL-26, GM-CSF, VEGF, LTα, IFNγ, IL-1β, IL-2, IL-8, IL-12p70, IL-13, IL-15, IL-16, and IL-17.

The non-significant results are shown in Appendix A (Figure A1 and Figure A2).

### 5.1. Analysis of All Patients’ Sera versus Healthy Controls

The results of the comparison between all patients’ cytokines sera levels versus healthy controls are presented in Table 2. This table shows only the statistically significant results about cytokines values altered in all three patients compared to healthy controls, which are CXCL-10 (*p*-value < 0.001), CCL-4 (*p*-value 0.044), and IL-6 (*p*-value 0.003). All of them were elevated in VL patients.

The conducted tests are Kruskal–Wallis in CCL-4 analysis, ANOVA in IL-6, and CXCL-10 analysis.

### 5.2. Sub-Analysis Stratified by Disease Stages: The Recovery Phase

The analysis of TNF-α and IL-7 in healthy controls versus all patients’ sera is reported in Table 3. These cytokines showed a statistically significant elevation only in the convalescent patient (Patient 1) compared to the healthy controls (*p*-values 0.022, <0.001, respectively). Conversely, TNF-α and IL-7 levels in the acute-phase patients were normal. In the first analysis (see Table 2), neither TNF-α nor IL-7 results were significantly altered.

The comparison was studied using Student’s *T*-test.

### 5.3. Sub-Analysis Stratified by Disease Stages: The Acute Phase

Table 4 enlists only the data concerning cytokines that were significantly altered in the acute-phase patients (Patients 2 and 3) compared to healthy controls. These cytokines are IL-1α (*p*-value 0.020), IL-5 (*p*-value 0.040), IL-10 (*p*-value < 0.001), and IL-12 (*p*-value 0.009).

IL-4 is comprehended in this analysis and enlisted in Table 4 even if it was not statistically significant as its *p*-value was superior to 0.05. IL-4 was allocated in this group as a *p*-value of 0.057 (Kruskal–Wallis test), even if not significant, might imply that the elevation of IL-4 could be linked to the acute manifestation of VL.

It is important to underline that in this sub-analysis concerning IL-1α, IL-4, IL-5, IL-10, and IL-12, the confrontation of all three patients (including Patient 1) versus healthy controls did not result in any statistical significance.

The comparison was studied using either the ANOVA test or the Kruskal–Wallis test. The relative *p*-value is enlisted in the tables; as we selected only the significant results, its value is always inferior to 0.05 except for IL-4, as mentioned above.

### 5.4. Interferon-γ(IFN-γ): The Missing Elevation

No significant result was obtained regarding the analysis of IFN-γ’s sera levels. In fact, while the healthy controls presented a medium value of IFN-γ of 4.22 pg/mL (interquartile range was 2.31–5.03), we found an IFN-γ value of 2.44, 5.89, and 9.00 pg/mL, respectively, in Patients 1 (convalescent-phase patients), 2, and 3 (acute-phase patients).

The comparison of the acute-phase patients versus the healthy controls by the Kruskal–Wallis’s test resulted in a *p*-value of 0.07, which is low, but not inferior to 0.05. Such a result represents the fact that this specific cytokine was higher in the acute-phase patients, but it cannot be considered statistically significant. This result was unexpected as interferon-γ plays such a pivotal role in the expression of a Th1 pattern of immune response during *Leishmania* spp. infection.

To obtain a graphic representation of the results, we elaborated 2D-dot-plot graphs reporting cytokines values in comparison. The confrontation of CXCL-10, CCL-4, and IL-6 sera levels between all our patients versus healthy controls is shown in Figure 3.

The sub-analysis stratified by disease stages is represented in Figure 4. It is a triple confrontation of the results concerning the acute phase cases (Patients 2 and 3), which are represented on the left, the healthy controls (represented in the middle), and the recovery-phase case (Patient 1) is represented on the right. While in the IL-α, IL-4, IL-5, IL-10, and IL-12 analyses the significant cytokine elevation relates to the acute-phase patients, TNF-α and IL-7 showed a higher level in the recovery patient compared to healthy controls.

## 6. Discussion

Clinically significant forms of leishmaniasis are rare because many of the exposed individuals harbor the parasite but never develop the disease, and only a small number of infected hosts exhibit symptoms of VL [32]. Most of the reported cases are represented by severely immunosuppressed individuals. The occurrence of disease in otherwise healthy patients in a non-endemic area is uncommon. Since our patients all had risk factors for previous *Leishmania* exposure, but there was no apparent trigger for parasite reactivation, we tried to analyze the possible causes of the development of VL in immunocompetent hosts.

### 6.1. COVID-19 Pandemic

The first thesis speculated about a possible role of the immune alterations due to the COVID-19 pandemic or its vaccines. Eventually, the SARS-CoV-2 infection and/or vaccination could have thrown off the immune system of our patients, leading to the reactivation of a previous silent *Leishmania* infection.

Rarely, some cases of *Leishmania* reactivation in immunocompetent patients have been described during the pandemic [33,34,35]. However, there are discrepancies on the subject all over the world as another recent review does not describe any cases of VL related to SARS-CoV-2 infection [36].

Our analysis assessed a significant elevation of IL-6 in all our patients, which is known to play a fundamental role in the physiopathology of severe SARS-CoV-2 infections [37,38]. Nevertheless, none of our patients presented a previous history of pneumonia or severe SARS-CoV-2 infection, which might be linked to a peak in IL-6 serum levels or in a cytokine storm severe enough to sustain a reactivation of a previously acquired parasitic infection. Also, the temporal correlation between SARS-CoV-2 infection and VL is unclear. All the patients contracted COVID-19 before presenting VL first symptoms but the time-window is very variable. Overall, it seems unwary to support this speculation unless new data emerge on the subject.

### 6.2. Epidemiology in Brescia

We estimated the crude incidence rate of VL by considering all the cases notified to the surveillance system of the Ministry of Health in people resident in the municipalities of the HPA of Brescia. In the population of the HPA of Brescia, only 7 cases of human VL were officially notified from 2005 to 2021, accounting for a crude incidence rate of 0.37 cases per million people per year.

Even if it is possible that the real VL incidence is higher, due to undiagnosed or unreported cases, leishmaniasis is a non-endemic disease in Brescia. Therefore, the hospitalization of three cases of human VL in less than three months is unusual. However, no correlation between the patients could be found: they live in different places, traveled in different countries, and none of their domestic animals were infected. Furthermore, no subsequent cases of VL came to our attention during the following two years (2022–2023). Overall, the hypothesis of an outbreak of leishmaniasis in Brescia cannot be advocated.

### 6.3. Cytokine Cascade

The main pro-inflammatory cytokines against VL, which mediate macrophage activation and intracellular parasite clearance, are represented by CXCL-10, IL-2, IL-8, IL-12, IL-15, IL-17, IL-22, and TNF-α. These molecules are responsible for naive T-cell polarization towards a Th1 phenotype with the production of IFN-γ, eventually killing the parasite.

On the other hand, a prevailing anti-inflammatory cytokine activation pattern, as in the Th-2 phenotype, may lead to the development of VL, since the result of its polarization consists of the inflammation shutdown, parasite survival, and consequent progression of the disease. The T-cell polarization towards a Th2 phenotype seems to be promoted by both the immune system (with a pivotal role of IL-10), and *Leishmania* amastigotes themselves [26].

The cytokine cascade activated in VL physiopathology is summarized in Figure 5.

A recently published review by Bohr et al. highlights that patients with active VL had higher plasma levels of IL-10, IL-4, IL-12, IFN-γ, and TNF-α than asymptomatic and cured subjects [26]. Another study, conducted by Costa et al., compared blood levels of pro- and anti-inflammatory cytokines in symptomatic, asymptomatic, treated, and uninfected patients, showing statistically significant higher blood levels of IFN-γ, TNF-α, IL-4, and IL-10 in the active disease [39].

The sera of our three patients were compared with healthy donors. Our analysis demonstrates statistically significant higher blood levels of CXCL-10, CCL-4, and IL-6 in all three patients. Moreover, IL-1α, IL-5, IL-10, and IL-12 levels were significantly elevated in the acute-phase patients. On the contrary, TNFα and IL-7 presented higher titers in our recovery-phase patient. These results are mostly consistent with other studies on the subject [7,8,11,26,39,40,41,42].

CXCL-10 blood levels of all the patients were significantly higher than healthy controls (*p*-value < 0.001). Its activity in VL pathogenesis is not entirely understood, but it seems to have an anti-parasite effect. Also, a study conducted in Brazil reports increased CXCL-10 levels in patients with active VL [41].

A higher expression of CCL-4 by infected macrophages has been previously described in active VL. It has been theorized that *Leishmania* itself promotes strategic expression of CCL-4 as it can induce the chemotaxis of TGF-β-producing cells, promoting Th2 activation and overall parasite survival [43]. CCL-4 was significantly elevated in all our patients (*p*-value 0.044).

IL-1 and IL-6 play a dual role in the pathogenesis of VL: IL-1 activates macrophages and stimulates parasite clearance, and IL-6 promotes inflammation and parasite killing. However, a recent review by Bohr et al. associates the elevation of IL-6 with the downregulation of the immune response and impairment of parasite clearance [26]. Both IL-1α and IL-6 were significantly altered in our patients’ profiles compared to healthy controls. IL-1α was elevated only in the acute-phase patients, suggesting an early activation in the immune response (*p*-value 0.020). Conversely, IL-6 was altered in all our patients (*p*-value 0.003). If the hypothesis of Bohr et al. is correct, and IL-6 played a role in the impairment of parasite clearance, this may help to explain VL development in immunocompetent hosts [26].

IL-4, IL-5, and IL-10, along with other CD4+ Th2-related cytokines, are associated with parasite survival inside the macrophages [26,44,45,46]. These cytokines were elevated in the patients affected by active VL (*p*-values 0.057, 0.040, and <0.001, respectively). Considering the fundamental role of IL-10 in the activation of the Th2 phenotype, its significant elevation represents a key step in the persistence of *Leishmania* in the infected hosts. Also, the low level of IL-10 in the recovering patient is consistent with parasite clearance [7,26,27,32].

A review by Costa et al. describes IL-12 as being higher in patients with active VL compared to carriers and cured individuals [39]. Lower levels of IL-12 might indicate the absence of an effective lymphocytic immune response [7,40,42]. In our study, IL-12 levels were consistently higher in the patients with active VL (*p*-value 0.009).

TNF-α is produced by macrophages to activate T-cells towards the Th1 pathway and enhance the macrophage activity for *Leishmania* clearance. High levels of TNFα are usually found in sera of patients affected by active disease, with a progressive decrease after the patient is exposed to treatment [26,47].

In our patients, a significant elevation of TNFα levels was demonstrated only in our recovery-phase patient (*p*-value 0.022). This result offers an ambiguous interpretation as it seems to be in contrast with the literature. However, a high TNFα titer in the convalescent phase, right after the completion of amphotericin B treatment may be an expression of the recent parasite clearance.

On the other hand, the low levels measured in the acute-phase patients, who had not been treated, could be related to a prevalent Th2 phenotype activation, resulting in parasite persistence. Further studies are needed to better assess the role and timing of the activation of TNFα and its responsibility in parasite persistence.

A similar result pertains to IL-7, which was significantly elevated only in the recovery-phase patient. The role of IL-7 is to stimulate hematopoiesis by triggering the proliferation of thymocytes, NK cells, and mature T cells. Also, IL-7, when combined with IFN-γ, promotes the synthesis and secretion of IL-6, TNFα, IL-1α, and IL-1β from monocytes to macrophages. In turn, these cytokines can induce the killing of amastigotes [26,48].

In the literature, an aggravation of leishmaniasis is described in murine models when IL-7 was prematurely risen in the initial phase of the parasite infection [49,50]. Consequently, the elevation of IL-7 in the recovery-phase patient may suggest its role in the latest immune response to VL. Another hypothesis might be that the absence of a proper IFN-γ peak elevation, which is a fundamental synergist in monocyte activation mediated by IL-7, can postpone its effects and, therefore, increase IL-7 levels during the recovery phase.

Also, the missing data are noteworthy, as we could not prove a significant elevation of IFN-γ, which should play a fundamental role in parasite clearance. In fact, IFN-γ levels were not significantly different in our population (*p*-value 0.07), although they were increased in the acute-phase patients. These data do not correlate with the literature, as IFN-γ is usually increased in active VL patients compared to healthy or treated individuals [26,27,39]. The missing substantial elevation of IFN-γ appears to be the main discrepancy between our study and the literature.

A possible explanation is that the lack of IFN-γ response represents itself as the output of a defective Th1 response, which could have favored the manifestation of VL. In fact, IFN-γ deficiencies are described in immunodeficient, autoimmune, and immune-dysregulated conditions [51,52,53].

However, the patients’ sera were not submitted to further studies to define whether they suffer from a deficit of IFN-γ. Therefore, further studies are needed to confirm this hypothesis.

### 6.4. Limitations of the Study

The first limitation of the study is due to VL diagnostic difficulties. The clinical presentation of the disease is indeed non-specific, and therefore it is often mistaken for hematological or oncological conditions. Even if an infectious diseases consultant is engaged in the differential diagnosis, it can be challenging to differentiate VL from other infections. Because of this complexity, the diagnosis of VL can be demonstrated sooner or later during the evolution of the disease stages. This happened with our patients too: their diagnoses did not all occur at the same time from the beginning of symptoms. It is possible that these discrepancies affected our measurements of cytokine levels since it was impossible to obtain all the patients’ sera in the same disease stage.

The second and main limitation of this study is the very small number of patients included. Even if VL is a very rare condition in the province of Brescia, and to observe three cases in less than three months is exceptional, it is overall a small population to analyze. A bigger number of cases would be needed in order not to affect the statistical data, or unduly infer causal relationships between the parameters in the study.

Further studies are needed to better describe the cytokine cascade in VL and assess its pathogenic roles.

## 7. Conclusions

Overall, the analysis of the cytokine activation pattern in our patients demonstrates a significant elevation of the main characters in the immune response against *Leishmania*.

The most interesting finding of our analysis is that most of the altered cytokines seem to be related to a pattern of imbalanced activation of the Th2 phenotype, which is known to support parasite survival. As the main limitation of our work is the small number of analyzed cases, it is impossible to assert with certainty whether these findings can fully explain the development of VL in immunocompetent and otherwise healthy individuals living in a non-endemic region.

What appears to be clear is that this rare and neglected parasite infection may present even in atypical patients. This calls for an improvement of research and training projects about *Leishmania* spp. to raise awareness of the disease among clinicians.

## Figures and Tables

**Figure 1 microorganisms-12-00394-f001:**
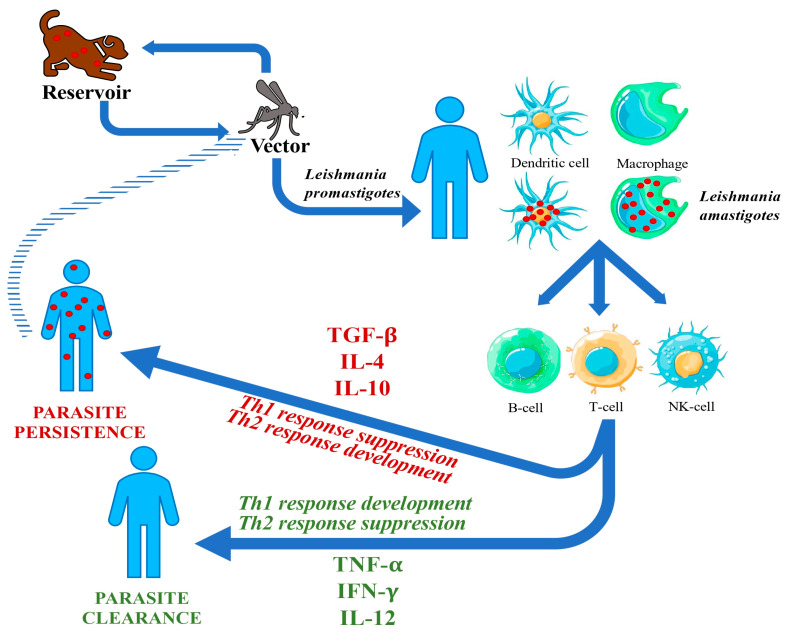
Leishmania biologic cycle and physiopathology of VL.

**Figure 2 microorganisms-12-00394-f002:**
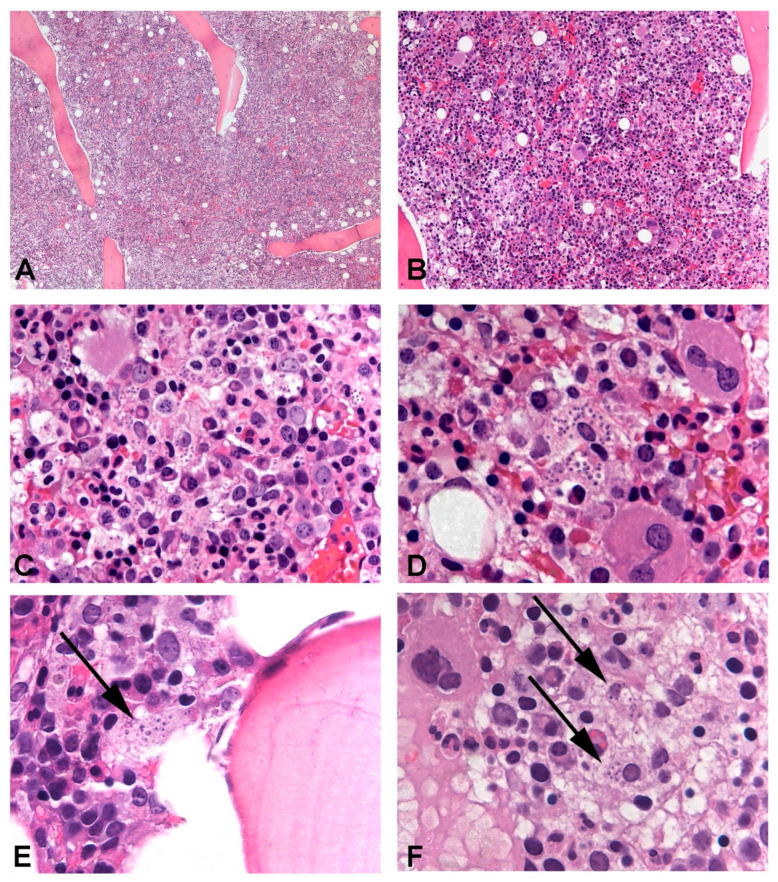
*Leishmania* amastigotes could be identified in the bone marrow trephine of all patients, in variable amounts. In Case 3, intertrabecular lacunae were hypercellular and numerous macrophages phagocyting significant numbers of amastigotes were evident and could be easily observed (**A**–**D**). In Cases 1 and 2, parasitic infection was more subtle with only a few amastigotes identifiable in rare macrophages ((**E**) and (**F**), respectively). In Case 1, cellularity was increased, while in Case 2, it was normal and hematopoietic progenitors of the three series displayed signs of abnormal maturation.

**Figure 3 microorganisms-12-00394-f003:**
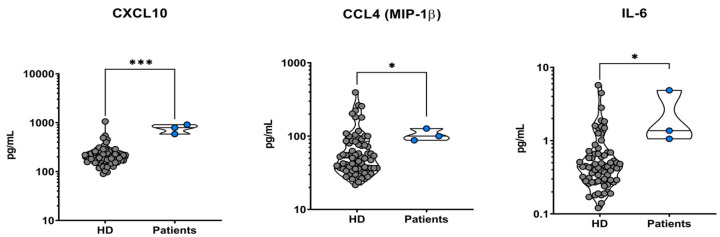
Two-dimensional dot-plot graphs of all patients (on the right of the graphs) vs. healthy donors of sera (on the left of the graphs, abbreviated as “HD”). All three patients presented elevated plasmatic levels of CXCL-10, CCL-4, and IL-6 when compared to the healthy controls (graphs from left to right). * indicates a statistically significant result (0.05 > *p*-value > 0.001), while *** evidences a strong statistical significance (*p*-value < 0.001).

**Figure 4 microorganisms-12-00394-f004:**
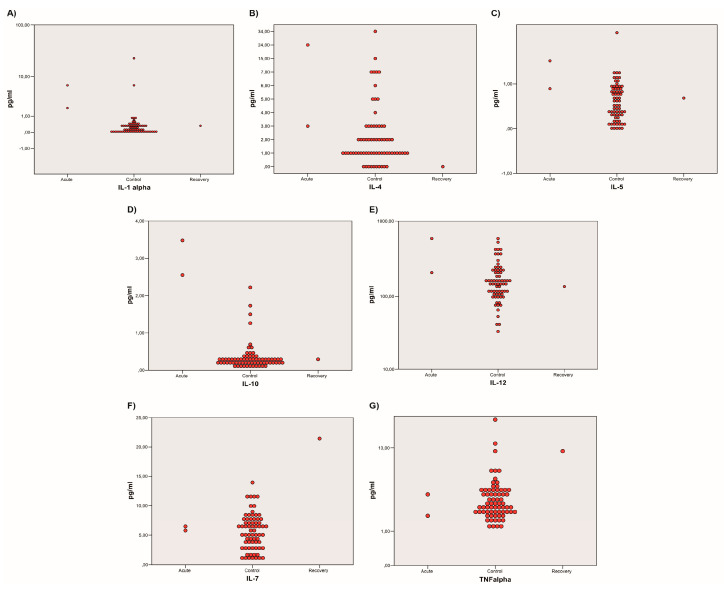
Two-dimensional dot-plot graphs comparing healthy controls to acute-phase or recovery-phase cases. Images (**A**–**E**) show that the acute-phase patients (on the left) presented higher concentrations of IL-1α, IL-4, IL-5, IL-10, and IL-12, respectively, while the recovery patient (on the right) can be assimilated to the healthy controls (in the middle). Images (**F**,**G**) represent TNF-α, and IL-7 that presented higher values in the convalescent patient (on the right) compared to the healthy controls (in the middle) and the acute-phase patients (on the right). All cytokines are expressed in pg/mL.

**Figure 5 microorganisms-12-00394-f005:**
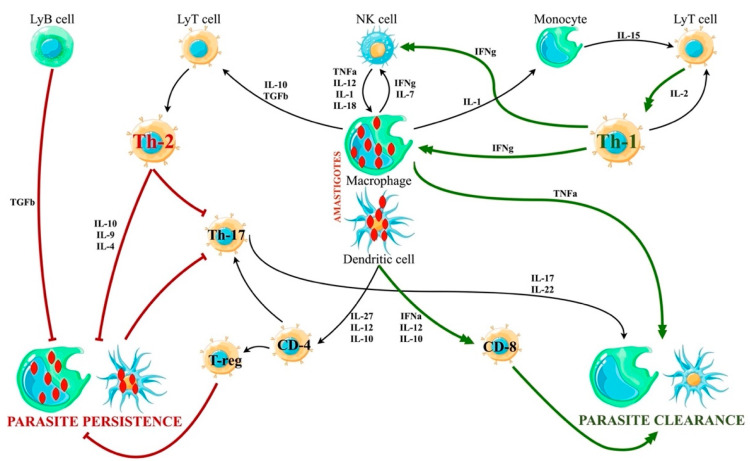
Cytokine cascade in VL. There are two main patterns of cytokine activation, leading to the expression of Th-1 or Th-2 phenotype of T-cell polarization, respectively. While the Th1 response empowers the parasite clearance, the Th2 activation facilitates the parasite persistence in macrophages and dendritic cells. The innate immune response activation usually consists of the expression of both Th1 and Th2 cells, but an unbalanced response may lead to the development of VL even in immunocompetent hosts.

**Table 1 microorganisms-12-00394-t001:** Summary of risk factors for exposure to *Leishmania* spp. and medical conditions known to tamper with the immune system response.

Risk Factors	Patient 1	Patient 2	Patient 3
Leishmania previous infection	No	No	No
Last travel in endemic area	Greece (October 2019)	Albania (August 2021)	Greece (August 2021)
Previous imported infections	None	None	None
Pets or animal exposure	Domestic dog	None	Domestic dog
Previous opportunistic infections	VZV, hHSV-1	None	VZV
HIV-antibodies	Negative	Negative	Negative
Autoimmunity markers	RF +	RF +, dCT +, ANA +, ↓ C3-C4	RF +, dCT +, ANA +

VZV: Varicella Zoster Virus; hHSV: human Herpes Simplex Virus; RF: Rheumatoid Factor; dCT: direct Coombs Test; ANA: anti-nuclear antibodies; +: positive; ↓: low sera levels.

**Table 2 microorganisms-12-00394-t002:** Enlistment of CXCL-10, CCL-4, and IL-6 values (pg/mL) of all patients and healthy controls (median value, interquartile range). The relative *p*-value is reported on the right column.

Cytokines	Healthy Controls	Patient 1	Patient 2	Patient 3	*p*-Value
CXCL-10	201, 168–248	584	914	792	<0.001
CCL-4	52, 36–87	88	100	127	0.044
IL-6	0.40, 0.28–0.60	1.06	4.90	1.40	0.003

**Table 3 microorganisms-12-00394-t003:** Enlistment of TNFα and IL-7 values (pg/mL) of all patients and healthy controls (median value, interquartile range). The relative *p*-value is reported in the right column.

Cytokines	Healthy Controls	Patient 1	Patient 2	Patient 3	*p*-Value
TNF-α	2.40, 1.27–7.82	9.01	1.80	3.21	0.022
IL-7	5.51, 3.05–7.51	21.43	6.04	6.41	<0.001

**Table 4 microorganisms-12-00394-t004:** Enlistment of IL-1α, IL-4, IL-5, IL-10, and IL-12 values (pg/mL) of all patients and healthy controls (median value, interquartile range). The relative *p*-value is reported in the right column.

Cytokines	Healthy Controls	Patient 1	Patient 2	Patient 3	*p*-Value
IL-1α	0.15, 0.04–0.3	0.3	1.8	6.5	0.020
IL-4	0.01, 0.01–0.03	0	0.03	0.24	0.057
IL-5	0.45, 0.20–0.80	0.58	0.86	1.86	0.040
IL-10	0.26, 0.2–0.3	0.3	3.5	2.5	<0.001
IL-12	150, 103–216	134	201	599	0.009

## Data Availability

The data presented in this study are available upon request from the corresponding author. The data are not publicly available due to privacy reasons.

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
