# Peer review of "Visceral Leishmaniasis in Immunocompetent Hosts in Brescia: A Case Series and Analysis of Cytokine Cascade"

_microorganisms, 2024, doi:10.3390/microorganisms12020394_

Round 1
Reviewer 1 Report (Previous Reviewer 1)
Comments and Suggestions for Authors
The authors present a case series and analysis of cytokines' cascade of Visceral leishmaniasis in immunocompetent hosts. The manuscript will be of interest to those working in leishmania infection. However, there are some points to take into account.
1.- If is possible, increase the quality of figures in appendix A.
2.- How the small number of analyzed cases affected the statistical data? Please discuss about it.
3.- How the time coursing with the disease, at the time of taking the sera, could affect the results? I mean the time transcurred when the disease was diagnosticated.
4.- Some parts in the discussion could be moved to introduction section, for example information related to figure 5.
Comments on the Quality of English LanguageMinor editing of English language required
Author Response
- If is possible, increase the quality of figures in appendix A.
Thank you for the suggestion. We improved the resolution of all the figures as much as we could. We hope that now the quality of the images is good enough to ease the graphs evaluation.
- How the small number of analyzed cases affected the statistical data? Please discuss about it.
We would like to thank you for raising this crucial point. The small number of analyzed cases is the main limitation of our study, as of course it affects the statistical data. We inserted a new little paragraph in the discussion to highlight the limitations of our study. You can read it from L432 to L448. We hope it will be enough to better explain this matter.
- How the time coursing with the disease, at the time of taking the sera, could affect the results? I mean the time transcurred when the disease was diagnosticated.
This is also a very interesting observation. As for the above point, we discussed it in the paragraph “Limitations of the study”.
- Some parts in the discussion could be moved to introduction section, for example information related to figure 5.
We would like to thank you for the appreciation and interest shown for our manuscript. Even if we agree that some parts of this section could be moved to the introduction, we feel that the paragraph “1.2 Physiopathology” already reports an extensive explanation of the role of the cytokine cascade in VL. Also, we chose not to split the discussion since we feel that for the reader it might be easier to focus on the alteration of the cytokine pattern found in our patients, following the explanation of their role in VL.
- Minor editing of English language required
We would like to thank you for the advice. We revised once again the manuscript to improve the editing of English language as you requested. We sincerely hope that the manuscript is now well written and easy to read.
Reviewer 2 Report (Previous Reviewer 2)
Comments and Suggestions for Authors
The manuscript of Mule et al. entitled Visceral leishmaniasis in immunocompetent hosts in Brescia: a case series and analysis of cytokines' cascade is the resubmitted version of the same manuscript describing three cases of visceral leishmaniasis and cytokines status study. The authors have added the parts of the results, tables and figures that were missing in the previous version.
Below are some comments and suggestions for the revealed issues:
L124: Usually RT-PCR stands for reverse transcription PCR, real-time PCR is abbreviated as qPCR.
L220: I suggest putting Appendix A as a separate Supplement Figure file.
L235: There is nothing about Student's t-test in the Materials and Methods, only about ANOVA t-test and Kruskal-Wallis.
Figure 3: What does HD mean?
Figures 4, A1, A2: The quality is not good (low resolution).
L443: First sentence is not complete?
Finally, the ethics approval statement is missing, which is a major issue for a manuscript describing patients.
Comments on the Quality of English LanguageEnglish language is just ok.
Author Response
- L124: Usually RT-PCR stands for reverse transcription PCR, real-time PCR is abbreviated as qPCR.
We would like to thank you for the advice. We have corrected the text as you suggested, you can now find the abbreviation “qPCR” in the last version of the manuscript.
- L220: I suggest putting Appendix A as a separate Supplement Figure file.
We would like to thank you for the advice, that we also received during the first review of the manuscript. We tried to do so during the submission process. However, the “supplementary materials” section does not allow to upload files of images such as jpeg.
- L235: There is nothing about Student's t-test in the Materials and Methods, only about ANOVA t-test and Kruskal-Wallis.
We apologize for this mistake which is now corrected in Materials and Methods section, at L209.
- Figure 3: What does HD mean?
The abbreviation HD indicates the healthy donors of sera, whose cytokine levels were compared to our patients’. Unfortunately, we did not clarify it in the main text. We apologize for this mistake which now has been corrected in the caption of the figure.
- Figures 4, A1, A2: The quality is not good (low resolution).
Thank you for the suggestion. We improved the resolution of all the figures as much as we could. We hope that now the quality of the images is good enough to ease the graphs evaluation.
- L443: First sentence is not complete?
We want to thank you very much for pointing out this typing error. We cancelled the half sentence.
- Finally, the ethics approval statement is missing, which is a major issue for a manuscript describing patients.
We would like to thank you for raising this crucial point. We previously discussed this matter with our Assistant Editor too. It is true that our manuscript does not include information regarding Ethics Committee or Institutional Review Board approval. However, Italian legislation does not oblige to get the approval of the Ethics Committee regarding case reports or case series. In fact, when we first submitted this manuscript, we contacted our Ethic Committee delegate who confirmed that, according to current Italian regulations, formal approval by the Territorial Ethics Committee is not required since our study is not a clinical trial but a case series. Nevertheless, our legislation imposes to obtain the informed consent of the patients to the publication. We collected all the informed consents as we reported at L210 “All subjects gave their informed consent for inclusion before they participated in the study. The study was conducted in accordance with the Declaration of Helsinki.” I hope our explanation may resolve this major issue.
Reviewer 3 Report (Previous Reviewer 4)
Comments and Suggestions for Authors
The authors address all the concers. Congratulation
Author Response
Dear reviewer,
We would like to thank you for the appreciation and interest shown for our manuscript. Thank for your consideration.
Best regards.
Round 2
Reviewer 2 Report (Previous Reviewer 2)
Comments and Suggestions for Authors
All issues have been addressed. The manuscript is ready to be accepted.
Comments on the Quality of English LanguageEnglish is okay.
This manuscript is a resubmission of an earlier submission. The following is a list of the peer review reports and author responses from that submission.
Round 1
Reviewer 1 Report
Comments and Suggestions for Authors
The authors present the analysis of cytokines’ cascade in three patients with visceral leishmaniasis. The idea is interesting; however, the manuscript has serious flaws, a lot of sections indicated have not description (maybe the file uploaded was incorrect), experimentally is limited, more evidence is needed to support the conclusions. Minor details, Table 1 is not cited in the text, the quality of images in Appendix A have to be improved.

Author Response
Dear reviewer,
We would like to thank you very much for your suggestions. Unfortunately, due to some malfunctioning in word to pdf conversion, the pdf version of the manuscript was incomplete. As suggested by the editor we will re-submit the complete text of the manuscript. We sincerely apologize for the inconvenient.
Regarding the missing citation of Table 1, thank you for the advice, we added it as you suggested.
Kind regards,
Alice Mulè
Reviewer 2 Report
Comments and Suggestions for Authors
The manuscript by A. Mule et al. entitled "Visceral leishmaniasis in immunocompetent hosts in Brescia: a case series and analysis of cytokines' cascade" describes three cases of leishmaniasis in an Italian city. To my great regret, the manuscript is incomplete. Obviously the authors did not finish the results, because there are only headings and no results. The mentioned Table 2 is also missing. For a finished text before the Results, I can only recommend to correct some flaws:
L44: There are four subgenera in the genus Leishmania, not two: Leishmania, Sauroleishmania, Mundinia, and Viannia.
L57: It may simply be VL.
L58, 60, etc.: Replace the period with a comma.
L79: Replace the comma with a period.
L121: Include how the RT-PCR was performed in the Materials and Methods. Which kit was used? Etc. in detail.
L166: Table 1 is not referenced in the text.
L172-174: What are CA, BM, SM?
L204: Epi Info and SPSS versions?
L214: Appendix must be in separate Supplement Figure file.
L221: ANOVA in capital letters.
There are no references to refs 32-52 in the manuscript.
Comments on the Quality of English LanguageSome minor editing would have benefited the text.
Author Response
Dear reviewer,
We would like to thank you very much for your suggestions. Unfortunately, due to some malfunctioning in word to pdf conversion, the pdf version of the manuscript was incomplete. As suggested by the editor we will re-submit the complete text of the manuscript. We sincerely apologize for the inconvenient.
Regarding all your other suggestions, thank you for the advice, we already corrected the text as you suggested.
Kind regards,
Alice Mulè
Reviewer 3 Report
Comments and Suggestions for Authors
The article "Visceral leishmaniasis in immunocompetent hosts in Brescia" presents a valuable but limited study due to its small sample size of three VL patients. This limitation may affect the robustness of the conclusions drawn. The absence of key sections in the manuscript (3.2, 3.3, 3.4, 4.1, 4.2, and 4.3) suggests a need for further elaboration. A constructive suggestion for the authors would be to consider reframing the manuscript as a case report that focuses more on detailed clinical and laboratory data, or to expand the patient cohort for a more comprehensive analysis, especially if continuing with cytokine profile exploration.
Author Response
Dear reviewer,
We would like to thank you very much for your suggestions. Unfortunately, due to some malfunctioning in word to pdf conversion, the pdf version of the manuscript was incomplete. As suggested by the editor we will re-submit the complete text of the manuscript. We sincerely apologize for the inconvenient.
Kind regards,
Alice Mulè
Reviewer 4 Report
Comments and Suggestions for Authors
First, I do not think the submitted manuscript is a complete one. You can see that the most of the result and discussion sections were empty. Thus, I would recommend "reject and resubmit after revision". In addition, the category of case report may be more appropriate for this submisison.
Author Response

(The authors gave the same response as above.)
